



# Influence of biomass burning from Southeast Asia at a high-altitude mountain receptor site in China

Jing Zheng[1], Min Hu[1,*], Zhuofei Du[1], Dongjie Shang[1], Zhaoheng Gong[2,a], Yanhong Qin[1], Jingyao Fang[1], Fangting Gu[1], Mengren Li[1], Jianfei Peng[1], Jie Li[3], Yuqia Zhang[3], Xiaofeng Huang[2], Lingyan He[2], Yusheng Wu[1], Song Guo[1]

[1]State Key Joint Laboratory of Environmental Simulation and Pollution Control, College of Environmental Sciences and Engineering, Peking University, Beijing, China

[2]Key Laboratory for Urban Habitat Environmental Science and Technology, School of Environment and Energy, Peking University Shenzhen Graduate School, Shenzhen, China

[3]State Key Laboratory of Atmospheric Boundary Layer Physics and Atmospheric Chemistry (LAPC), Nansen-Zhu International Research Center (NZC), Institute of Atmospheric Physics, Chinese Academy of Sciences, Beijing, China

*Correspondence to: minhu@pku.edu.cn

[a] Now at John A. Paulson School of Engineering and Applied Sciences, Harvard University, Cambridge, Massachusetts, 02138, United States

**Abstract**

Highly time-resolved in-situ measurements of airborne particles were made at Mt. Yulong (3410 m above sea level) on the southeastern edge of the Tibetan Plateau in China from 20 March to 14 April in 2015. Detailed chemical composition was measured by a high-resolution time-of-flight aerosol mass spectrometer together with other online instruments. Average mass concentration of the submicron particles ($PM_1$) was $5.7 \pm 5.4$ μg m$^{-3}$ during the field campaign, ranging from 0.1 μg m$^{-3}$ up to 33.3 μg m$^{-3}$. Organic aerosol (OA) was the dominant component in $PM_1$, with a fraction of 68%. Three OA factors, i.e., biomass-burning organic aerosol (BBOA), biomass-burning-influenced oxygenated organic aerosol (OOA-BB) and oxygenated organic aerosol (OOA), were resolved using positive matrix factorization analysis. The two oxygenated OA factors accounted for 87% of the total OA mass. Three biomass burning events were identified by examining the enhancement of black carbon concentrations and the $f_{60}$ (the ratio of the signal at m/z 60 from the mass spectrum to the total signal of OA). Back trajectories of air masses and satellite fire map data were integrated to identify the biomass burning locations and pollutants transport. The western air mass from Southeast Asia with active biomass burning activities transported large amount of air pollutants, resulting in elevated organic concentrations up to 4-fold higher than that





of the background condition. This study at Mt. Yulong characterizes the tropospheric background
aerosols of the Tibetan Plateau during pre-monsoon season, and provides clear evidence that the
southeastern edge of the Tibetan Plateau is affected by transport of anthropogenic aerosols from
Southeast Asia.
**1 Introduction**
Aerosols play an important role in the radiative balance in earth's atmosphere, with its radiative forcing
still having large uncertainties (IPCC, 2013). Biomass burning emission is one of the dominant sources
of atmospheric particles (von Schneidemesser et al., 2015), contributing up to 90% of the primary organic
aerosol in the global scale (Bond et al., 2004) and more than half of the total organic aerosol mass in
areas with significant biomass burning influences (e.g. Yangtze River Delta region in China, and Indian
Peninsula) (Zhang et al., 2015; Engling and Gelencser, 2010). Given the long atmospheric lifetime of
aerosols, even remote areas can sometimes be influenced by the transportation of air pollutants from
areas with active biomass burnings (Bougiatioti et al., 2014). In terms of the deterioration of air quality
and climate change in those remote areas, great scientific interest has arisen focusing on the impacts on
biomass burning (Lau et al., 2010; Qian et al., 2011).
The Tibetan Plateau is the largest and highest plateau in the world, and is often regarded as the "Third
Pole". It is surrounded by a ring of high-elevated mountain ranges, which were considered as blocks for
transportations of air pollutants from its vicinity (Wang and French, 1994). Since this vast land has a
relatively low population density with minor anthropogenic influences, the Tibetan Plateau has been
considered as a natural background of the Eurasian continent (Ming et al., 2010; Wan et al., 2015). In
recent years, studies have presented convincing evidences for the transport route of air pollutants
climbing over the Himalayas, especially during pre-monsoon season, coinciding with the annual
intensive fire season in South and Southeast Asia (Streets et al., 2003; Marinoni et al., 2010; Cong et al.,
2015b). A westerly dry circulation helps to build up the smoke plume against the Himalayan ridges,
elevating to 3-5 km in altitude (Bonasoni et al., 2010; Xia et al., 2011). Subsequently, downward glacier
wind of local mountain breeze circulation brings biomass burning related air pollutants down to the
mountain valley (Cong et al., 2015b; Lüthi et al., 2015).





A host of studies based on field campaigns have amassed an impressive amount of information
describing the biomass burning influence on different areas of the Tibetan Plateau (Decesari et al., 2010;
Zhao et al., 2013; Xu et al., 2015). Those studies were mostly approached by analyzing the temporal and
spatial variations of atmospheric composition based on filter measurements. The strong correlation of
carbonaceous aerosol with biomass burning tracers $K^+$ and levoglucosan pointed out the origins of
aerosols (Cong et al., 2015a). Biomass burning organic aerosol (BBOA) was also found to be a major
fraction of organic aerosol (OA), with a 15% contribution to the total OA mass (Du et al., 2015). Xu et
al. (2013) and You et al. (2016) also presented convincing evidences about biomass burning impacts by
analyzing chemical components in glaciers collected in the Tibetan Plateau. Most of previous studies
were based on offline analysis using filter or glacier samples, which were limited to low time resolution,
making it difficult to follow the aging process of biomass burning aerosol. Thus in-situ measurements of
aerosol chemical characterization with high time resolution are needed, so as to have a deep
understanding of the sources and evolution of the particulate matter.
In this study, the influence of biomass burning from Southeast Asia on the Tibetan Plateau will be
analyzed. The results can serve as inputs or constraints for global climate model simulations. By
examining the aerosol properties as a function of chemical compositions at Mt. Yulong at the southeastern
edge of the Tibetan Plateau, this study sheds light on the evolution processes of OA. Positive matrix
factorization analysis was conducted to resolve different sources of OA, and characterize the influence
of biomass burning from Southeast Asia transported over long distances to the Tibetan Plateau
background environment during pre-monsoon season.
**2 Method**
**2.1 Site description and meteorological conditions during the campaign**
In this study, we conducted an intensive observation at the site on Mt. Yulong (27.2N, 100.2E), with an
altitude of 3410 m a.s.l., northwestern Yunnan Province, China (Fig.1). Since Mt. Yulong is lying in the
transition zone extending from the low altitudes of the Yunnan Plateau (~ 3000 a.s.l.) to the high altitude
of the Tibetan Plateau (~ 5000 a.s.l.), it is on the transport route of pollutants from Southeast Asia to
inland China, making it to be an ideal site to observe the influence of regional and long-range transport





of polluted air masses. This station is a member of the National Atmospheric Watch Network coordinated
by the Chinese Environmental Monitoring Center. The famous tourist attraction Lijiang Old Town locates
more than 36 km away and 1000 m lower than the elevation of the station. The observation period was
conducted during the pre-monsoon season of the Tibetan Plateau, from 22 March to 14 April 2015,
corresponding to the annual biomass burning seasons in Southeast Asia. Since the season was cold with
sparse visitors in Lijiang old city, the influence of local emissions from residents and visitors remained
low compared with other seasons.
**2.2 Measurements and data processing**
A high resolution time-of-flight aerosol mass spectrometer (AMS hereafter) was deployed to measure
the highly time-resolved chemical composition of sub-micron, non-refractory aerosols (Table S1). The
standard operation procedures of the AMS has been described in detail in Canagaratna et al. (2007).
During the field operation, the AMS alternated between V and W modes, allowing the acquirements of
averaged chemical compositions of the non-refractory particles, as well as high resolution mass spectrum
of organics. The detection limits (DL) of organic, sulfate, nitrate, ammonium and chloride were 0.07,
0.004, 0.003, 0.005 and 0.01 $\mu$g m$^{-3}$, respectively. During most time of the campaign, the mass
concentrations of chloride were below its DL, and including it would lower the total signal to noise ratio,
therefore it is omitted from the analysis.
The AMS data was analyzed using the standard AMS data analysis software, i.e., SQUIRREL (version
1.57) for unit resolution mass spectrum data, and PIKA (version 1.16) for high resolution mass spectra
data. Calibrations of the AMS on flow rate and ionization efficiency were conducted each week. To
account for the particle loss due to the bounce of particles on the vaporizer, collection efficiencies were
calculated and applied for data correction based on the method described by Middlebrook et al. (2012).
The high resolution organic aerosol spectra were further apportioned to different sources by positive
matrix factorization (PMF) analysis (Paatero and Tapper, 1994; Ulbrich et al., 2009). The solution was
validated by the characteristics of resolved mass spectra, as well as the comparison of temporal variations
between each factor and external species (e.g. acetonitrile).
Other online instruments were also deployed at the site (Table S1). A scanning mobility particle sizer



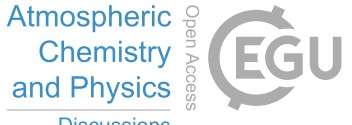

(SMPS) was used to measure particle number size distribution for particle mobility diameters ranging
from 3 to 780 nm, with a time resolution of 5 min. An Aethalometer was deployed to measure the aerosol
light absorption coefficients $\sigma_{ap}$ at its seven wavelengths, ranging from 370 to 950 nm. Black carbon (BC)
concentration is determined by $\sigma_{ap}$ at 880 nm using the default mass attenuation cross sections of 16.6
$m^2 \, g^{-1}$ (Fröhlich et al., 2015). Acetonitrile was measured by a gas chromatographer with mass
spectrometer and flame ionization detectors (GC–MS/FID) with a time resolution of 1 hour. Technical
details of this self-made instrument were described elsewhere (Wang et al., 2016).
Meteorological parameters, including relative humidity, temperature, wind direction and wind speed,
are continuously monitored on the site during the campaign. The low temperature (5 °C for the whole
campaign average) and heavy snow eliminated the influence of biogenic emissions to this site during the
campaign.
**2.3 Back trajectory analysis and fire maps**
To explore the influence of regional biomass burning activities on aerosol properties during the
campaign, the Weather Research and Forecasting (WRF) model (version 3.61) was used to investigate
the meteorological conditions and to compute trajectories of air mass arriving at Mt. Yulong. 48-h back
trajectories were calculated every 6 hours from March 22 to April 14, using a starting height at 600 m
above the ground level of the site.
Active fire points were obtained from the Fire Information for Resource Management System
(FIRMS), which is provided by the Moderate Resolution Imaging Spectroradiometer (MODIS) satellite
(https://firms.modaps.eosdis.nasa.gov/firemap/, last accessed on Aug. 26, 2016).

**3 Results**
**3.1 Concentrations and chemical compositions of submicron aerosols**
The time series of submicron aerosol compositions as well as meteorological conditions are shown in
Fig.2. The average PM$_1$ concentration was $5.7 \pm 5.4$ µg m$^{-3}$, with a range of 0.1 - 33 µg m$^{-3}$. This result
was similar to previous observations at the Northern Tibetan Plateau, where Du et al. (2015) reported an



average PM$_1$ concentration of 11.4 µg m$^{-3}$ in the autumn of 2013, and Xu et al. (2014a) reported an annual
average PM$_{2.5}$ concentration of 9.5 µg m$^{-3}$ from 2006 to 2007. The averaged PM$_1$ concentration was
much lower than those measured at urban and downwind sites of China (e.g., (Huang et al., 2013; Xu et
al., 2014b), but was three times higher than the 1.7 µg m$^{-3}$ at a background site in Europe in March 2004
(Sjogren et al., 2008), and ten times higher than at the same background site in Europe in the spring of
2013 (Fröhlich et al., 2015). These huge differences indicate that anthropogenic pollutions in Southeast
Asia may have resulted in the elevation of aerosol concentrations to levels above the natural background
level.
Averaged aerosol composition of PM$_1$ is shown in the pie chart (Fig.3(a)). The PM$_1$ chemical
composition was dominated by organic components, which accounted for 68%, followed by sulfate
(14%). The minor contribution of nitrate to PM$_1$ (4%) can be explained by the lack of nearby
anthropogenic sources for precursors (e.g., HONO, N$_2$O$_5$) (Du et al., 2015). This result presents a similar
picture as those observed at remote sites in the northern hemisphere (Zhang et al., 2011), as well as at a
high altitude site in Europe (Ripoll et al., 2015). Compared with urban or regional areas in China, where
secondary inorganic species including sulfate, nitrate and ammonium typically contribute to over one
half of the total mass concentrations, the result at this site is quite unique (Huang et al., 2010; Huang et
al., 2012; Xu et al., 2014b).
Fig. 3(b) shows the relative contribution of major chemical components as a function of PM$_1$ mass
concentrations, as well as the probability density of PM$_1$ mass loading. PM$_1$ concentrations below 5 µg
m$^{-3}$ showed highest probability (68%). The fraction of organics and BC increase slightly with the
increasing of PM$_1$ concentrations, showing that they were the main contributors to the pollution episode
in Mt. Yulong.
The PM$_1$ components did not show distinct diel variations, but remained relatively constant during the
whole day, as shown in Fig.3(c). This is similar to the findings at the Puy-de-Dôme station in central
France, and the Montsec station in western Mediterranean Basin (Freney et al., 2011; Ripoll et al., 2015).
Strong long-range transport of air mass with few local emissions could blur the diel cycles, since the air-
mass transportations occurred regardless of the local time of the day.





**3.2 Characterization of organic aerosol**
**3.2.1 Elemental compositions of organic aerosol**
The elemental composition were calculated from high resolution mass spectra of organics obtained by
AMS, using the method developed by Canagaratna et al. (2015). Compared with the previous method
(Aiken et al., 2007; 2008), the ratio of O/C and H/C are typically increased by 20% and 7%, respectively.
Bulk OA was mainly composed of carbon and oxygen, with minor contributions from hydrogen and
nitrogen, and had an average molecular formula of $C_1H_{1.4}O_{1.1}N_{0.04}$. The fragments of organics were
grouped into five types according to the existence of C, H, O or N atoms. $C_xH_y^+$ were only 21% of the
total organic signal, while the oxygen fraction ($C_xH_yO_z^+$) accounted for 68% of the total OA, which is
higher than those measured at urban and downwind site (30-41%) (Huang et al., 2011; Sun et al., 2011;
Hu et al., 2013). The average OM/OC and O/C ratios for the whole campaign were 2.63 and 1.11,
respectively, and were similar to those measured in the north eastern region of the Tibetan Plateau
(OM/OC 2.75, O/C 1.16) (Xu et al., 2015). These results are slightly higher than the elemental ratios
measured at another remote site (OM/OC: 2.4, O/C: 0.9) in the eastern Mediterranean (Bougiatioti et al.,
2014), probably due to the mixture of free troposphere aerosol after a long time of processing before
arriving at this high altitude site. The extremely high value of OM/OC reflects the highly oxidized nature
of OA in the Tibetan Plateau.
**3.2.2 Source apportionment of organic aerosol**
PMF analysis was performed to investigate the sources of OA measured at Mt. Yulong. Three factors
were resolved, including a biomass burning organic aerosol (BBOA), an oxygenated biomass-burning-
influenced organic aerosol (OOA-BB), and an oxygenated organic aerosol (OOA). Details of the PMF
analysis can be found in the supplement. The mass spectra of the three factors are shown in Fig. 4. The
time series of the three factors and an external species (acetonitrile) are plotted in Fig. 5.
3.2.2.1 BBOA
BBOA has been frequently identified in previous studies at urban and regional sites (Zhang et al.,
2011). The mass spectrum of BBOA has a notable contribution from m/z 60 (mainly $C_2H_4O_2^+$,
contributing 3.1% of the total mass spectra), which is from fragmentation of levoglucason. As shown in
Table 1, the mass spectrum correlates well with the samples from an aircraft measurement above a large





forest fire (Brito et al., 2014), and well with the samples from biomass burning simulation system in the
laboratory (He et al., 2010). BBOA has an O/C ratio of 0.37, presenting a similar level to previous studies
(Aiken et al., 2008; He et al., 2010). The time series of BBOA correlates very well with $K^+$ based on filter
analysis (Pearson R=0.95). The factor was also confirmed to be BBOA by its strong correlation with
acetonitrile, which is a gas phase tracer for biomass burning.

The average concentration of BBOA was 0.5 μg m$^{-3}$ for the whole campaign, accounting for 13% of

the total OA mass, with a maximum contribution at 61% (Fig.6 (a)). The spikes in the time series of
BBOA indicate that a fraction of BBOA was contributed by primary sources nearby, possibly occasional
biomass burning activities for domestic heating and cooking. The increasing fraction of BBOA as a
function of total OA concentrations points to contributions from biomass burning activities during the
pollution episode (Fig.6 (b)).
3.2.2.2 OOA-BB

The mass spectrum of OOA-BB factor was dominated by $C_xH_yO_z^+$ fragments, especially org29

(CHO$^+$), org43 (C$_2$H$_3$O$^+$) and org44 (CO$_2^+$). The spectrum of OOA-BB in this study well correlated with
aged BBOA obtained 3 hours downwind of a forest fire (Brito et al., 2014) (Pearson R=0.97). It is
qualitatively similar to published OOA-BB spectra from aged BB plumes in China during the harvest
seasons (Zhang et al., 2015), and also presented many similarities to those of OOA2-BBOA resolved in
the metropolitan area of Paris (Crippa et al., 2013).

The average concentration of OOA-BB was 0.9 μg m$^{-3}$ for the whole campaign, accounting for 22%

of the total OA mass. Compared with BBOA measured near sources, OOA-BB shows a higher
oxygenated degree, with an O/C of 0.85, and a lower fraction of m/z 60 (0.6%), as a result of the oxidation
of primary levoglucosan-type species after long periods of atmospheric processing (Jolleys et al., 2015).
As the plumes originated from Southeast Asia were measured at a distance of several hundred kilometers
downwind, emissions would have undergone substantial aging prior to sampling. The aging process
includes both the gas-phase oxidation of semi volatile species from biomass burning sources and
heterogeneous or homogeneous reactions of existing particles during long-range transport (Bougiatioti
et al., 2014). The time series of OOA-BB and BBOA yield modest correlations with BC (Pearson R=0.62
and 0.65). If we focus on the total biomass burning related organic aerosols (OOA-BB + BBOA), the R





value for its correlation with BC would increase to 0.76, indicating biomass burning related OA
originated from the same source as BC.
3.2.2.3 OOA
OOA is described as highly oxidized, aged particles formed after long-range transportation and
processing. The mass spectral properties of OOA are defined by having a dominant peak at m/z 44
(mainly $CO_2^+$) and other ions of $C_xH_yO_z^+$. The highly oxidized nature of OOA is also reflected by its
high O/C ratio of 1.45. The mass spectrum of OOA resembles that of more oxidized OOA (MO-OOA)
in Beijing well (Hu et al., 2016) (Pearson R=0.69).
OOA has an average concentration of 2.6 μg m$^{-3}$, accounting for 65% of the total OA mass. Unlike
previous studies at urban or regional sites (Jimenez et al., 2009; Li et al., 2015; Hu et al., 2016), the time
series of OOA did not agree well with that of sulfate (Pearson R=0.32), which was also the case at the
puy-de-Dome research station (1465 m a.s.l.) (Freney et al., 2011). The low Pearson correlation value
can be partially explained by the extremely high concentration of OOA formed from the oxidation of
organics emitted by biomass burning activities during the first week of the campaign. For the rest of
campaign, the correlation value for sulfate with respect to OOA factor increases to 0.77, which is
consistent with previous studies.
As shown in Fig. 6(a), the two OOA factors (OOA-BB and OOA) were very abundant, with a
predominantly contribution of 87% to the total OA mass. This is consistent with the high oxygen level
in the total OA. During 80% of the observation period, OA concentrations were lower than 5 μg m$^{-3}$,
with strong contributions from secondary organic aerosols (OOA and OOA-BB) (Fig.6 (b)). This
indicates that the background site was predominated by organic aerosols formed through regional
transportation.
**4 Discussion**
**4.1 Identification of biomass burning events**
Enhanced BC concentrations were used to help identifying periods influenced by biomass burning
plumes (Bougiatioti et al., 2014). The BC concentration of 85 ng m$^{-3}$ was taken as the background
concentration at this site. It is the average concentration observed in the beginning of April (1 April to 4





April), when the strong wind scavenged pollutants of the whole region. Back trajectory and fire maps
illustrate that the dominant air mass for this period was from north India with minor biomass burning
activities (see Fig.7(d)). This concentration is consistent with the two-year averaged background level
measured at Southern Himalayas (Marinoni et al., 2010), and comparable to the lowest BC concentrations
found over the southeastern Tibetan Plateau in the pre-monsoon season (Engling et al., 2011).
During the sampling period, three episodes were identified as being influenced by biomass burning,
with the following criteria satisfied. It includes: (a) Back trajectory analysis shows a uniform source
region; (b) Fire map shows fire spots in the region during the episode; (c) BC concentrations were higher
than the background level of 85 ng m$^{-3}$ determined above. One long-lasting and strong episode was from
22 March to 30 March. The air mass arrived at the site during this period was from south part of Myanmar,
and covered active biomass burning areas (see Fig.7(a)). As shown on the fire map, the site may also be
influenced by wildfires in the vicinity. Two less intense events were observed on 5 - 6 April and 11 - 12
April, with slightly elevated BC concentrations. During the third event (11 - 12 April), the site
experienced heavy snow. The back trajectory shows that air mass to this region was transported from
regions with few fire spots. The enhanced BC concentration was probably emitted by biomass burning
activities nearby for domestic heating and cooking.
These three biomass burning events were further validated by the increase of the fraction of biomass
burning tracers, $f_{60}$, calculated as the ratio of the signal at m/z 60 to the total OA signal. During the first
and second events, the average $f_{60}$ were 0.98% and 0.61%, respectively. These values were much lower
than the $f_{60}$ of 1.4% during the third event, which was influenced by fires in the vicinity. This showed the
decay of $f_{60}$ in ambient plumes transported from sources to the receptor site. During the clean episode,
the $f_{60}$ decreased to about 0.4%, indicating minor biomass burning influence (Cubison et al., 2011).
The box plot (Fig.8) shows the concentrations of different chemical components of biomass burning
events and background conditions. The aerosol concentrations corresponding to the background
condition is highlighted by light gray. Organic aerosols use the left axis while other species use the right
axis, since concentration of OA were much higher than others. Aerosols corresponding to biomass
burning events were at high concentrations. The concentrations of organic aerosol during three biomass
burning events were 10, 4 and 6 folders higher than that of the background condition. During the first





event, due to co-occurrence of biomass burning activities in the vicinity together with the long-range
transport of biomass burning plume, the concentration of BC reached 14 times higher than that of the
background condition. All species remained at low and sustained background concentrations during the
clean episode, with an average $PM_1$ concentration of 1.2 µg m$^{-3}$.
**4.2 Characteristic of three Biomass burning events**

The comparison of OA fractions of different biomass burning events is shown in Fig.9. Since the air

masses arriving at Mt. Yulong during the second event was transported from active biomass burning
areas in Myanmar within 48h, most of the freshly emitted BBOA were processed and transformed to
more oxidized OA, with OOA and OOA-BB together accounting for 90% on average of the total organic
mass. In contrast, the fraction of BBOA had strong enhancement during the third event, reaching 23%.
It is consistent with the previously mentioned identification that the biomass burning plumes were mainly
from residential heating nearby, which could emitted large amount of fresh BBOA.

The aging and/or mixing processes of different biomass burning plumes are further characterized in

terms of the $f_{44}$ vs. $f_{60}$ triangle plot (Cubison et al., 2011). $f_{44}$, similarly defined to $f_{60}$ as the ratio of the
signal at m/z 44 to the total OA signal, is used here as an indicator of atmospheric aging, since OA and
their gas phase precursor evolve in the atmosphere by becoming increasingly oxidized with higher $CO_2^+$
fraction (Jimenez et al., 2009; Ng et al., 2010). BBOA can be clearly distinguished from oxidized OA in
the triangle plot. With the aging process of biomass burning plumes, OA evolved toward higher $f_{44}$ and
lower $f_{60}$, and gained more similar signature with OOA.

The OA clusters of three biomass burning events are shown clearly in the $f_{44}$-$f_{60}$ triangle plot (Fig.10).

The OA clusters of the first and third events both present OA peaks with high $f_{60}$ values, since the site
was possibly influenced by residential heating in the surrounding regions during these two episodes. The
OA cluster of the second event present more similar oxidative properties to OOA and OOA-BB, due to
loss of biomass burning marker through aging process during transport.
**5 Conclusions**

During the pre-monsoon season the aerosol evolution was explored at a high altitude receptor site on

Mt. Yulong (3410 m a.s.l.) in the Tibetan Plateau in Southwestern China. The average concentration of

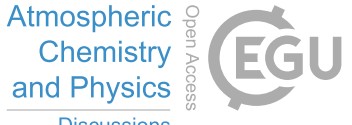

$PM_1$ was 5.7 µg m$^{-3}$, which was far below that measured in urban and suburban as well as regional sites
of China. The carbonaceous species were very abundant in $PM_1$, with an average contribution of 68%,
followed by sulfate (14%) and BC (9%). This high altitude mountain site is suitable for tracing the
influence of pollution plumes transported from the large areas of Southeast Asia.
Using PMF analysis, organic aerosol was resolved into three factors, BBOA, OOA-BB and OOA.
OOA-BB formed after atmospheric process of BBOA during long-range transport. The two oxygenated
OA factors (OOA and OOA-BB) accounted for 87% of the total OA, showing the highly oxidized nature
of aerosol at the Mt. Yulong.
Different types of biomass burning events were identified by examining organic tracer in mass profiles
and BC concentrations.   The origins of biomass burning plumes were verified by analyzing the back
trajectories of air mass as well as fire maps. Elevated $PM_1$ concentrations due to the transport of air
pollutants from active biomass burning areas in Southeast Asia were observed.   Domestic heating
activity also had interference on the background condition of Mt. Yulong.
This study provides clear evidence on the influence of the transport of pollutants emitted by biomass
burning activity in Southeast Asia on the southeastern edge of the Tibetan Plateau in China. The chemical
characteristics of aerosols observed by in situ measurement can serve as inputs for model validations of
aerosol-cloud processes and long-range transports. This study also highlights the impact of
anthropogenic emissions to the pristine region of the Tibetan Plateau, which may influence global climate.

**Acknowledgements**
This study was supported by the National Natural Science Foundation of China (91544214,
21190052 and 41121004) and the China Ministry of Environmental Protection's Special Funds for
Scientific Research on Public Welfare (20130916). We also thank China National Environmental
Monitoring Center for the support to the field campaign.

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



**Table1. Comparison between mass spectra of different OA with reference spectra.**

|  | Reference spectra | Pearson Correlation Coefficient | | | Citation |
|---|---|---|---|---|---|
|  |  | BBOA | OOA-BB | OOA |  |
| Ambient measurement | 900m above fire | 0.91 | 0.56 | 0.34 | Brito et al. (2014) |
|  | 3h downwind | 0.51 | 0.97 | 0.91 | Brito et al. (2014) |
|  | MO-OOA | 0.69 | 0.86 | 0.69 | Hu et al. (2016) |
|  | BBOA | 0.85 | 0.38 | 0.11 | Hu et al. (2016) |
| Laboratory simulation | wood of pin | 0.91 | 0.61 | 0.42 | He et al. (2010) |
|  | rice straw | 0.94 | 0.6 | 0.36 | He et al. (2010) |



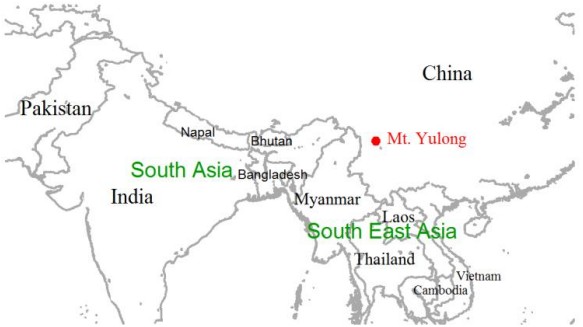


**Fig.1 The location of the sampling site at Mt. Yulong (27.2N 100.2E, 3410 m a.s.l.).**




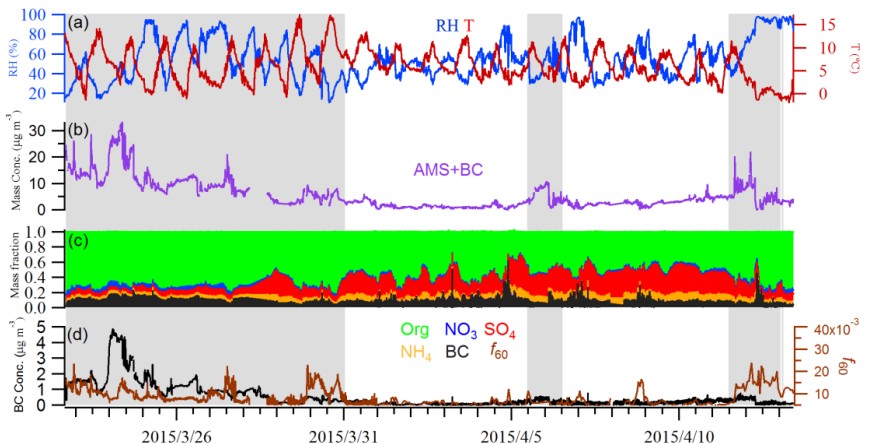


**Fig.2 Time series of (a) relative humidity and temperature; (b) total mass concentrations from AMS plus**
**black carbon (c) mass fractions of different chemical species; (d) concentrations of black carbon and $f_{60}$. The**
**gray background denotes three biomass burning events (identified in Section 4.1).**

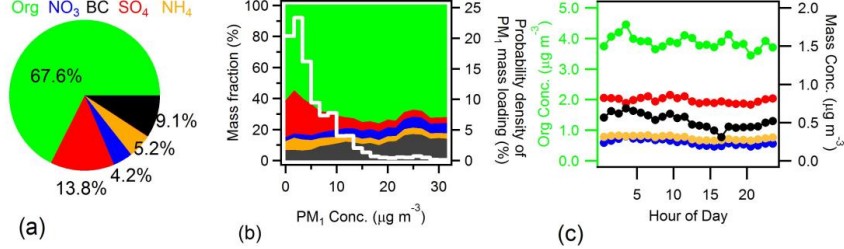


**Fig.3 (a) average chemical composition of the whole campaign; (b) the mass fractions of PM$_1$ species as a**
**function of PM$_1$ mass loading (left axis), with the white line representing the probability density of PM$_1$ mass**
**loadings (right axis); (c) the diel cycle of different species, with the left axis for organics, and the right axis for**
**the rest components.**





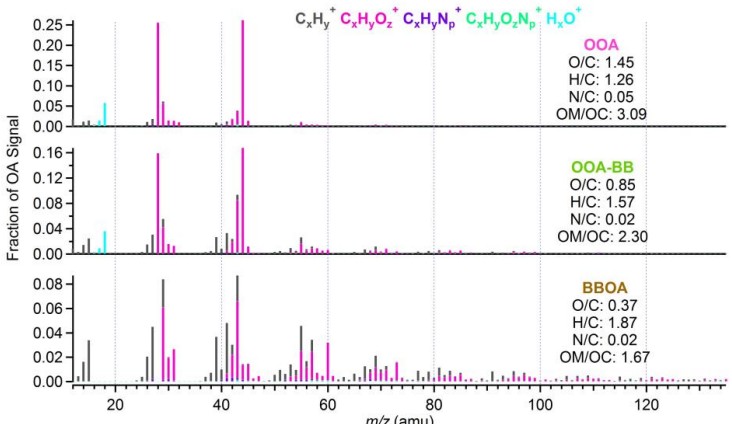


**Fig.4 The mass spectra of each factor resolved by PMF, together with atomic ratios of each factor.**

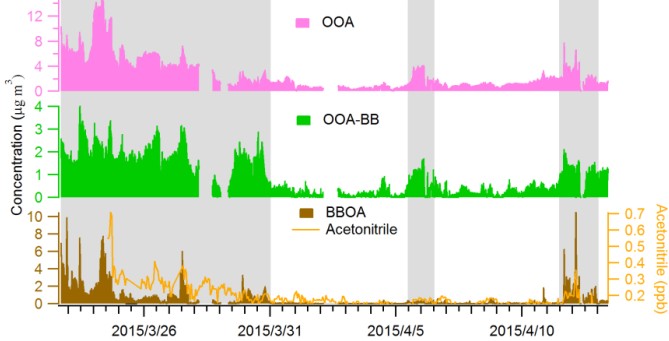


**Fig.5 The time series of three OA factors resolved by PMF, together with acetonitrile, a gas phase tracer for**

**biomass burning. The grey background areas denote the biomass burning events (identified in Section 4.1).**

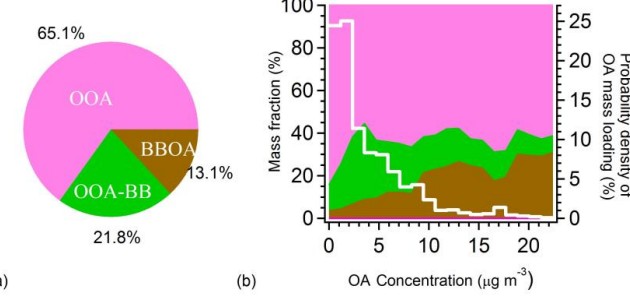

(a)                    (b)
**Fig.6 (a) contribution of each factor to the total OA mass; (b) fractions of OA factor (left axis) and**
**probability density of OA concentration (white line, right axis) as a function of OA mass loading.**





**Fig.7 Occurrence of wildfire derived from MODIS images and back trajectories (blue lines) from WRF**

**model (a) first biomass burning event: March 22$^{nd}$ – 30$^{th}$; (b) second biomass burning event: April 5$^{th}$ – 6$^{th}$;**
**(c) third biomass burning event: April 11$^{th}$ – 12$^{th}$; (d) background: April 1$^{st}$ – 4$^{th}$.**

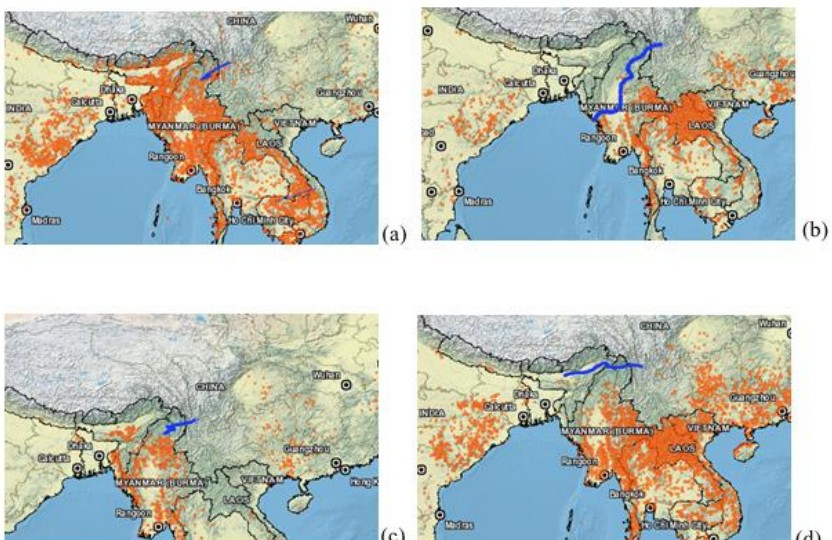


**Fig.8 Comparison of chemical compositions between three biomass burning events (BB1, BB2, BB3) and**
**background conditions (BG, highlighted by light gray color). Boxes denote median, 25th and 75th percentiles;**
**whiskers represent 5th and 95th percentiles; crosses represent mean values.**




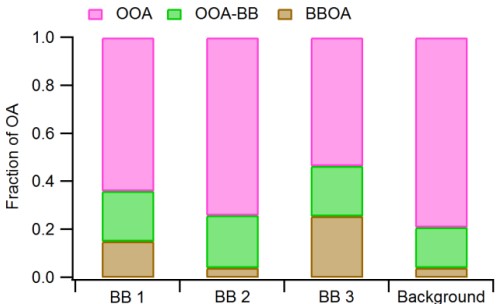


**Fig.9 The relative contribution of different types of OA during three biomass burning events and**
**background condition.**

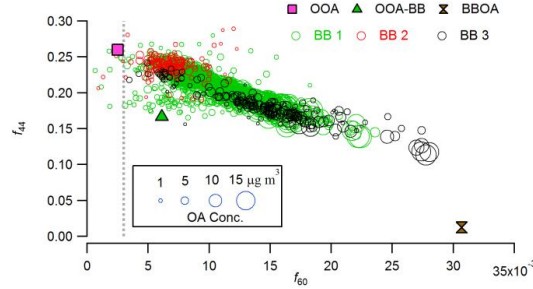


**Fig.10 $f_{44}$ as a function of $f_{60}$ ($f_{44}$ vs. $f_{60}$ triangle plot) of the three biomass burning events, sized by OA**
**concentration.**