# Peer review of "Influence of biomass burning from South Asia at a high-altitude mountain receptor site in China"

_Atmospheric Chemistry and Physics, 2016_

## Referee Comment (RC1) · Anonymous Referee #2 · 17 Mar 2017

The manuscript represents the results obtained by using a high-resolution time-of-flight aerosol mass spectrometer to monitor the chemical composition of submicron non-refractory aerosol at a high-altitude mountainous site at the edge of the Tibetan Plateau in China. The dominant component was found to be organic aerosol, with a contribution of 68%. Organic aerosol (OA) was found to be highly oxygenated and oxidized, depicting the secondary and processed nature of it. When deconvolving the organic aerosol to different factors, the two oxygenated factors, one originating from the processing of biomass burning and one highly oxidized, accounted for the 87% of the organic aerosol. The rest was identified as biomass burning organic aerosol (BBOA) originating from long-range transport both from neighboring countries (Myanmar) but also

from nearby activities, exhibiting different degrees of processing according to its prove-nance. Concentrations of OA during the biomass burning influence were significantly higher than those during background conditions, denoting the important influence of long-range transport pollution at remote areas of the globe. Overall the manuscript is interesting and easy to follow. It focuses on an area of the globe that is not extensively studied, adding value to the reported results. Additionally, it demonstrates the importance of the influence from polluted areas to remote background sites situated away from local sources. Nevertheless there are some important details missing and more thorough discussion should be made in specific sections. Other than that the paper can be recommended for publication after addressing the issues listed below.

Specific comments:

1) It is listed in the measurements section (2.2) that a 7-wavelegth aethalometer was also collocated at the site. Based on the approach of Sandradewi et al. (2008) a source apportionment of the black carbon (BC) measurements could be effectuated for the distinction between a wood burning and a fossil fuel component. This could significantly increase the confidence in the identification of the biomass burning events and could possibly improve the correlations between the time series of OOA-BB and BBOA with BC. Since the measurements are there, it is a pity not to take advantage of all the information.

2) In the same section (2.2) it is stated that a Scanning Mobility Particle Sizer (SMPS) was used to measure the size distributions during the measurement campaign. How did the integrated mass from the SMPS compare to the added masses from the AMS and BC measurements?

Other minor comments:

L36: . . .with their radiative forcing..

L65: . . .convincing evidence. . ..

L66: Most of the previous studies. . .

L84: . . .making it an ideal site. . . .(delete "to be")

L157: . . .showed the highest probability. . .

L167: . . .elemental composition was calculated. . .

L173: . . .oxygenated fraction (instead of "oxygen")

L207: It is stated that the OOA-BB of the study correlated well with aged BBOA from 3 hours downwind of a forest fire. Yet, in L214 it is noted that m/z 60 is reduced after long periods of atmospheric processing. This is somehow contradictory as 3 hours seems quite quick. Similar to the observations of the current study, Minguillón et al. (2015) identified a quite oxygenated factor (named OOAm from mix) which was partially made up from processed BBOA quite quickly and that explained the fact that the time series of OOAm tracked that of the BBOA. Therefore it is suggested to rephrase L214 as to reflect the quick nature of this processing.

L254: Replace "It includes" maybe with "Among other"

L257: . . .from the south part. . .

L272: OA are represented by the left axis while other species are "represented" by the right axis.

L275: . . .were 10, 4 and 6-fold higher. . .

L278: . . .remained low. . .

L282: Air mass or air masses (as mentioned before in the text)? Be consistent.

L287: . . .which could emit . . .

---

## Referee Comment (RC2) · Anonymous Referee #1 · 20 Mar 2017

This paper reports the chemical composition of submicron non-refractory aerosol at Mt. Yulong, a mountainous site at the southeast edge of the Tibetan Plateau in China, by using a high-resolution time-of-flight aerosol mass spectrometer, with other instruments such as aethalometer, SMPS, GC-MS/FID. The aerosol at Mt. Yulong was dominant by organic aerosol, following by sulfate, BC. The manuscript paid more attention to deconvolve the organic aerosol into three factors, i.e. biomass-burning organic aerosol (BBOA), BBOA, biomass-burning-influenced oxygenated organic aerosol (OOA-BB) and oxygenated organic aerosol (OOA). OOA-BB and OOA accounted for about 87% of organic aerosol, which suggests that the OA is highly oxygenated in this remote site. This study also points out that the southeastern edge of the Tibetan Plateau is affected

by transport of anthropogenic aerosols from Southeast Asia. This work will add some understanding of the influence of biomass burning on remote site of Tibetan Plateau. The manuscript could be accepted for publication after revision.

There are some specific comments for authors: 1ãĂĄ The authors mentioned in abstract "...is affected by transport of anthropogenic aerosols from Southeast Asia." In fact, according to the back-trajectory, the aerosols were from South Asia, not Southeast Asia. Please check. 2ãĂĄ In section "Measurements and data processing", the authors should add more information about AMS size calibration, what the CE value were used, what the AMS data time resolution for data collection and later analysis were. 3ãĂĄ In Fig.S1, the mass concentration of PM1 measured by SMPS and that of AMS plus Aethalometer has compared. The authors mentioned the the estimated composition-dependent density was used. Please add what the value of aerosol density used? 4ãĂĄ Biomass burning event 2 is different with the others, the contribution of BBOA is comparable to the background period. It would be better to have more evidence to support the explaination. 5ãĂĄ Pearson Correlation Coefficient has mentioned many times in this manuscript, as we know, this value is related with the data points, please make them clear. 6ãĂĄ Line 80 Mt. Yulong (27.2oN, 100.2oE), 7ãĂĄ Line 113, the authors mentioned "particle number size distribution for particle mobility diameters ranging from 3 to 780 nm, In Table S1 just list 3081DMA and CPC 3775? Do you use two SMPS in parallel or not? 8ãĂĄ Line 304 "The carbonaceous species were very abundant in...", Carbonaceous species should include BC. Please rewrite this sentence. 9ãĂĄ Line 567, the authors mentioned the axis for different species in the text, but not in the figure caption, please make the figure self-readable.

---

## Author Comment (AC1) · 14 Apr 2017

We are thankful to the reviewers for their thoughtful and constructive comments on our study. The manuscript has been revised accordingly. Listed below is our point-to-point response to each comment.

General comments:

The manuscript represents the results obtained by using a high-resolution time-of-flight aerosol mass spectrometer to monitor the chemical composition of submicron non-refractory aerosol at a high-altitude mountainous site at the edge of the Tibetan Plateau in China. The dominant component was found to be organic aerosol, with a contribution of 68%. Organic aerosol (OA) was found to be highly oxygenated and oxidized, depicting the secondary and processed nature of it. When deconvolving the organic aerosol to different factors, the two oxygenated factors, one originating from the processing of biomass burning and one highly oxidized, accounted for the 87% of the organic aerosol. The rest was identified as biomass burning organic aerosol (BBOA) originating from long-range transport both from neighboring countries (Myanmar) but also from nearby activities, exhibiting different degrees of processing according to its provenance. Concentrations of OA during the biomass burning influence were significantly higher than those during background conditions, denoting the important influence of long-range transport pollution at remote areas of the globe. Overall the manuscript is interesting and easy to follow. It focuses on an area of the globe that is not extensively studied, adding value to the reported results. Additionally, it demonstrates the importance of the influence from polluted areas to remote background sites situated away from local sources. Nevertheless there are some important details missing and more thorough discussion should be made in specific sections. Other than that the paper can be recommended for publication after addressing the issues listed below.

Specific comments:

Q1: It is listed in the measurements section (2.2) that a 7-wavelegth aethalometer was also collocated at the site. Based on the approach of Sandradewi et al. (2008) a source apportionment of the black carbon (BC) measurements could be effectuated for the distinction between a wood burning and a fossil fuel component. This could significantly increase the confidence in the identification of the biomass burning events and could possibly improve the correlations between the time series of OOA-BB and BBOA with BC. Since the measurements are there, it is a pity not to take advantage of all the information.

Reply: We thanks the reviewer for the comment. It would be nice to be able to separate the fossil fuel component from the wood burning component, however in our study, the approach of Sandradewi et al. (2008) could not be a valid method. The approach needs accurate parameterization of $\alpha TR$ for traffic and $\alpha WB$ for wood burning, which can only be obtained when measurements were done near the sources. In the case of our study, as the BC undergoes the transportation process, it gets coated, thus its AAE decreases (Chung et al., 2012). Most of the BC

measured at the site was from transportation, thus gives us big uncertainties on the αTR and αWB values. The measurement was taken at a high mountain site, with the nearest city located 20 km to the south, and more than 1000 m below its altitude. The air masses were mainly from the west or southwest of the site, covering mostly remote areas. Thus the traffic source should play a minor role in this study.

Q2: In the same section (2.2) it is stated that a Scanning Mobility Particle Sizer (SMPS) was used to measure the size distributions during the measurement campaign. How did the integrated mass from the SMPS compare to the added masses from the AMS and BC measurements?

Reply: We thank the reviewer for this question. We have done this comparison, and is shown in Figure S1 in the supplementary materials part of this manuscript.

Other minor comments:

L36: with their radiative forcing ..

Reply: Revised.

L65: convincing evidence ..

Reply: Revised.

L66: Most of the previous studies ..

Reply: Revised.

L84: making it an ideal site ..(delete "to be")

Reply: Revised.

L157: showed the highest probability ..

Reply: Revised.

L167: elemental composition was calculated ..

Reply: Revised.

L173: oxygenated fraction (instead of "oxygen")

Reply: Revised.

L207: It is stated that the OOA-BB of the study correlated well with aged BBOA from 3 hours downwind of a forest fire. Yet, in L214 it is noted that m/z 60 is reduced after long periods of atmospheric processing. This is somehow contradictory as 3 hours seems quite quick. Similar to the observations of the current study, Minguillón et al. (2015) identified a quite oxygenated factor

(named OOAm from mix) which was partially made up from processed BBOA quite quickly and that explained the fact that the time series of OOA tracked that of the BBOA. Therefore it is suggested to rephrase L214 as to reflect the quick nature of this processing.

Reply: We thank the reviewer for the suggestion. We have revised Line181 as follow.

"Compared with BBOA measured near sources, OOA-BB shows a higher oxygenated degree, with an O/C of 0.85, and a lower fraction of m/z 60 (0.6%), as a result of the oxidation of primary levoglucosan-type species (Jolleys et al., 2015). This oxidation process can be quick in elevation of the oxidation state, as also identified in another study by Minguillón et al. (2015)."

L254: Replace "It includes" maybe with "Among other"

Reply: We thank the reviewer for the suggestion. The sentence is revised as follow: "with the following criteria satisfied, i.e., (a) back…"

L257: from the south part ..

Reply: Revised.

L272: OA are represented by the left axis while other species are "represented" by the right axis.

Reply: Revised.

L275: were 10, 4 and 6-fold higher ..

Reply: Revised.

L278: remained low ..

Reply: Revised.

L282: Air mass or air masses (as mentioned before in the text)? Be consistent.

Reply: Revised.

L287: which could emit ..

Reply: Revised.

Reference

Chung, C. E., Kim, S. W., Lee, M., Yoon, S. C., and Lee, S.: Carbonaceous aerosol AAE inferred from in-situ aerosol measurements at the Gosan ABC super site, and the implications for brown carbon aerosol, Atmospheric Chemistry and Physics, 12, 6173-6184, 10.5194/acp-12-6173-2012, 2012.

Sandradewi, J., Prévôt, A. S. H., Szidat, S., Perron, N., Alfarra, M. R., Lanz, V. A., Weingartner, E., and Baltensperger, U.: Using Aerosol Light Absorption Measurements for the Quantitative Determination of Wood Burning and Traffic Emission Contributions to Particulate Matter, Environ. Sci. Technol., 42, 3316–3323, 2008.

---

## Author Comment (AC2) · 14 Apr 2017

We are thankful to the reviewers for their thoughtful and constructive comments on our study. The manuscript has been revised accordingly. Listed below is our point-to-point response to each comment.

General comments:

This paper reports the chemical composition of submicron non-refractory aerosol at Mt. Yulong, a mountainous site at the southeast edge of the Tibetan Plateau in China, by using a high-resolution time-of-flight aerosol mass spectrometer, with other instruments such as aethalometer, SMPS, GC-MS/FID. The aerosol at Mt. Yulong was dominant by organic aerosol, following by sulfate, BC. The manuscript paid more attention to deconvolve the organic aerosol into three factors, i.e. biomass-burning organic aerosol (BBOA), BBOA, biomass-burning-influenced oxygenated organic aerosol (OOA-BB) and oxygenated organic aerosol (OOA). OOA-BB and OOA accounted for about 87% of organic aerosol, which suggests that the OA is highly oxygenated in this remote site. This study also points out that the southeastern edge of the Tibetan Plateau is affected by transport of anthropogenic aerosols from Southeast Asia. This work will add some understanding of the influence of biomass burning on remote site of Tibetan Plateau. The manuscript could be accepted for publication after revision.

Specific comments:

Q1: The authors mentioned in abstract": : :is affected by transport of anthropogenic aerosols from Southeast Asia." In fact, according to the back-trajectory, the aerosols were from South Asia, not Southeast Asia. Please check.

Reply: We thank the reviewer for the comment. We have revised "Southeast Asia" to "South Asia" in the title and in the manuscript.

Q2: In section "Measurements and data processing", the authors should add more information about AMS size calibration, what the CE value were used, what the AMS data time resolution for data collection and later analysis were.

Reply: We thank the reviewer for the comment. We have revised section 2.2 "Measurements and data processing" with the information added in Line 82 and Line 91 as follow.

"The time resolution was 5 min for AMS measurement, with 2.5 min in V mode to obtain mass concentration, and 2.5 min in W mode for HR mass spectrum of organics."

"In this study, the collection efficiencies varied from 0.5 to 0.9."

Q3:In Fig.S1, the mass concentration of $PM_1$ measured by SMPS and that of AMS plus Aethalometer has compared. The authors mentioned the estimated composition-dependent density was used. Please add what the value of aerosol density used?

Reply: We thank the reviewer for the question. We have added this information regarding aerosol

densities in the supplementary materials part by adding the following sentence in SI Line 19:

"The aerosol density used was 1.4 g cm$^{-3}$ for organics (Gysel et al., 2007), 1.75 g cm$^{-3}$ for sulfate, nitrate and ammonium, and 1.80 g cm$^{-3}$ for black carbon as suggested by Middlebrook et al. (2012)."

Q4:Biomass burning event 2 is different with the others, the contribution of BBOA is comparable to the background period. It would be better to have more evidence to support the explanation.

Reply: We thank the reviewer for the comment. We have added the following sentences in Line 242 – Line 243:

"Although the ratio of BBOA to the total OA during this event has a similar level with the background level, the mass concentrations of both OA and BBOA are much elevated than the background level."

Q5: Pearson Correlation Coefficient has mentioned many times in this manuscript, as we know, this value is related with the data points, please make them clear.

Reply: We thank the reviewer for the comment. We have added the number of data points at places where we mentioned the Pearson R values.

Q6: Line 80 Mt. Yulong (27.2oN, 100.2oE)

Reply: Revised.

Q7: the authors mentioned "particle number size distribution for particle mobility diameters ranging from 3 to 780 nm, In Table S1 just list 3081DMA and CPC 3775? Do you use two SMPS in parallel or not?

Reply: We thank the reviewer for the comment. We have modified in Line 96: "… 3 – 780 nm…" to "… 15 – 760 nm…", and have revised Table S1 as the follow.

"SMPS (3081-DMA and 3022-CPC)"

Q8: The carbonaceous species were very abundant in: : :", Carbonaceous species should include BC. Please rewrite this sentence.

Reply: We thank the reviewer for the comment. It is revised in Line 259 as follow: The carbonaceous species (OA+BC) were very abundant in $PM_1$, with an average contribution of 77%, followed by sulfate (14%) and ammonium (5%).

Q9: Line 567, the authors mentioned the axis for different species in the text, but not in the figure caption, please make the figure self-readable.

Reply: We thank the reviewer for the suggestion. It is revised in Line 491: OA are represented by the left axis while other species are represented by the right axis.

Reference

Gysel, M., Crosier, J., Topping, D. O., Whitehead, J. D., Bower, K. N., Cubison, M. J., I.Williams, P., Flynn, M. J., McFiggans, G. B., and Coe, H.: Closure study between chemical composition and hygroscopic growth of aerosol particles during TORCH2, Atmos. Chem. Phys., 7, 6131-6144, 10.5194/acp-7-6131-2007, 2007.

Middlebrook, A. M., Bahreini, R., Jimenez, J. L., and Canagaratna, M. R.: Evaluation of Composition-Dependent Collection Efficiencies for the Aerodyne Aerosol Mass Spectrometer using Field Data, Aerosol Sci. Technol., 46, 258-271, 10.1080/02786826.2011.620041, 2012.